# Interferon-α-Induced Dendritic Cells Generated with Human Platelet Lysate Exhibit Elevated Antigen Presenting Ability to Cytotoxic T Lymphocytes

**DOI:** 10.3390/vaccines9010010

**Published:** 2020-12-24

**Authors:** Ippei Date, Terutsugu Koya, Takuya Sakamoto, Misa Togi, Haruhiko Kawaguchi, Asuka Watanabe, Tomohisa Kato, Shigetaka Shimodaira

**Affiliations:** 1Department of Regenerative Medicine, Kanazawa Medical University, Uchinada, Kahoku 920-0293, Japan; dat-pey@kanazawa-med.ac.jp (I.D.); koya@kanazawa-med.ac.jp (T.K.); taku0731@kanazawa-med.ac.jp (T.S.); m-togi@kanazawa-med.ac.jp (M.T.); harukawa@kanazawa-med.ac.jp (H.K.); asuka-w@kanazawa-med.ac.jp (A.W.); 2Center for Regenerative Medicine, Kanazawa Medical University Hospital, Uchinada, Kahoku 920-0293, Japan; 3Medical Research Institute, Kanazawa Medical University, Uchinada, Kahoku 920-0293, Japan; tkato@kanazawa-med.ac.jp

**Keywords:** immunotherapy, dendritic cells, vaccine, human platelet lysate, tumor-associated antigens, cytotoxic T lymphocyte, endocytosis, proteolytic activity

## Abstract

Given the recent advancements of immune checkpoint inhibitors, there is considerable interest in cancer immunotherapy provided through dendritic cell (DC)-based vaccination. Although many studies have been conducted to determine the potency of DC vaccines against cancer, the clinical outcomes are not yet optimal, and further improvement is necessary. In this study, we evaluated the potential ability of human platelet lysate (HPL) to produce interferon-α-induced DCs (IFN-DCs). In the presence of HPL, IFN-DCs (HPL-IFN-DCs) displayed high viability, yield, and purity. Furthermore, HPL-IFN-DCs displayed increased CD14, CD56, and CCR7 expressions compared with IFN-DCs produced without HPL; HPL-IFN-DCs induced an extremely higher number of antigen-specific cytotoxic T lymphocytes (CTLs) than IFN-DCs, which was evaluated with a human leukocyte antigen (HLA)-restricted melanoma antigen recognized by T cells 1 (MART-1) peptide. Additionally, the endocytic and proteolytic activities of HPL-IFN-DCs were increased. Cytokine production of interleukin (IL)-6, IL-10, and tumor necrosis factor (TNF)-α was also elevated in HPL-IFN-DCs, which may account for the enhanced CTL, endocytic, and proteolytic activities. Our findings suggest that ex-vivo-generated HPL-IFN-DCs are a novel monocyte-derived type of DC with high endocytic and proteolytic activities, thus highlighting a unique strategy for DC-based immunotherapies.

## 1. Introduction

Dendritic cells (DCs) are antigen-presenting cells (APCs) that play a central role in immune acquisition [1]. DCs are widely expressed on body surfaces and in tissues characterized by distinct phenotypes, origins, receptors, and functions [2]. Human DCs in the blood can be divided into conventional DCs (cDCs) and plasmacytoid DCs (pDCs). cDCs are involved in the stimulation of CD4^+^ and CD8^+^ T cells whereas pDCs produce type 1 interferon (IFN) in response to a virus [3]. Recently, single-cell RNA sequencing has been used to genetically classify human blood DCs into six new subtypes [4]. Conversely, Langerhans cells and dermal DCs are found in human skin. Dermal DCs are classified into CD14^+^ DCs, CD141^+^ DCs, and CD1c^+^ DCs, which have been investigated for their functional relationship, similar to blood DCs [5]. Furthermore, DCs play a role in anti-cancer immunity and the prevention of infectious agents. Of the DC subsets, blood CD141^+^ DCs have a high antigenic cross-presentation capability, and thus, contribute to anti-tumor immunity [6]. Apart from distinct DC subsets in vivo, there are monocyte-derived DCs that can be produced ex vivo [7]. DCs take up tumor-associated antigens released from tumor cells and migrate to the lymph nodes, where they present these peptides to naïve CD8^+^ T cells through the human leukocyte antigen (HLA). CD8^+^ T cells primed by DCs become cytotoxic T lymphocytes (CTLs), which are recruited to the tumor microenvironment to kill tumor cells [8].

Since the 1990s, monocyte-derived DCs loaded with tumor-specific peptides such as Wilms tumor 1 (WT1) have been utilized in clinical studies and trials as vaccines against a variety of cancers [9]. In recent years, clinical trials have been conducted that combine these DCs with chemotherapy, such as S-1, and immune checkpoint inhibitors, such as nivolumab and ipilimumab [10,11,12]. Moreover, there has been significant interest in clinical studies on personalized vaccines, which use DC vaccines with autologous whole tumor lysates to amplify T cells that target somatic mutation-derived neoantigens [13].

Per the standardized manufacturing protocol of DC vaccines, leukapheresis is performed on cancer patients to collect monocyte-rich peripheral blood at the first step. Then, adherent cells isolated from the peripheral blood are stimulated with granulocyte macrophage colony-stimulating factor (GM-CSF) and interleukin (IL)-4 to differentiate into immature DCs. Various methods can be used to induce the maturation of these immature DCs, including the addition of cytokines such as tumor necrosis factor (TNF)-α; adjuvants such as lipopolysaccharide (LPS), a component of the outer membrane of the cell wall of gram-negative bacteria; and a cocktail of the streptococcal preparations OK-432 and prostaglandin E_2_ (PGE_2_) [14,15]. DC vaccines manufactured via these methods are administered intradermally to the axilla and/or groin, and DCs that take up tumor antigens migrate to the lymph nodes and activate T cells.

The revised IFN-DC protocol has been demonstrated in research development to be a superior protocol for manufacturing DC vaccines. Compared with IL-4 DCs, IFN-DCs generated from human monocytes using GM-CSF and interferon (IFN)-α can induce differentiation in a shorter period of time [16]. Furthermore, compared with IL-4 DCs, IFN-DCs have been reported to be more capable of cross-presenting antigens to CD8^+^ T cells via major histocompatibility complex class I molecules after processing them into peptides [17]. Clinical trials have been conducted using IFN-DC vaccines for medullary thyroid tumor, advanced melanoma, and follicular lymphoma [18,19,20]. Thus, the combination of additives is particularly important because of the different characteristics of DCs generated by cytokines and adjuvants.

There are other problems associated with the manufacturing of DCs for clinical trials, such as the use of animal resource reagents that have a potential risk to infect cells with animal pathogens. Since safety is essential when applying DC vaccine therapy, the use of xeno-free reagents is recommended in addition to other cell therapies [21]. Recently, human platelet lysate (HPL) has been reported as an alternative to fetal bovine serum (FBS) for various cell types and cellular therapies; specifically, HPL has been used with mesenchymal stem cells to treat steroid-refractory acute graft-versus-host-disease (GVHD), endothelial colony-forming progenitor cells for transplantation into patients with cardiac diseases, and endothelial cells for tissue repair [22,23,24]. HPL contains a variety of factors, including cell proliferation-related (e.g., platelet-derived growth factor; PDGF, transforming growth factor-β; TGF-β), cell adhesion-related (e.g., sICAM-1, sVCAM-1), and inflammatory response-related (e.g., sCD40L) factors, that have favorable effectiveness in cell culture [25]. The use of HPL as an alternative to FBS when culturing IL-4 DCs has also been reported; however, there were no noticeable differences in endocytic activity, viability, and the expression of cell surface markers between the use of FBS and HPL [26]. Various interactions between monocytes and platelets have been reported, and the effects of platelets and their contents on monocytes and DCs have received much attention [27]. Therefore, we conducted IFN-DC vaccine manufacturing using a xeno-free medium and HPL with a low adherent dish, with the results indicating a unique phenotype and function of the subsequent HPL-IFN-DCs.

## 2. Materials and Methods

### 2.1. Ethics

We performed DC generations in compliance with the Good Gene, Cellular, and Tissue-based Products Manufacturing Practice (GCTP). This study was evaluated and approved by the Ethical Committee of Kanazawa Medical University (approval number G131 and I489). All investigations were performed according to the Declaration of Helsinki, and all cells were obtained from patients after receiving the appropriate informed consent.

### 2.2. Cell Separation and Culture

#### 2.2.1. Preparation of Peripheral Blood Mononuclear Cells (PBMCs)

PBMC-rich fractions were collected by a Spectra Optia^®^ cell separator (Terumo BCT, Inc., Tokyo, Japan) using leukapheresis. Mononuclear cell-enriched fractions were separated from an apheresis product via density gradient centrifugation using Ficoll-Plaque Premium (Global Life Sciences Solutions USA LLC, Marlborough, MA, USA), and platelets were washed out with phosphate buffered saline (PBS).

#### 2.2.2. Generation of IFN-DCs

PBMCs from the patients were suspended in serum-free medium DCO-K (Nissui Pharmaceutical Co., Ltd., Tokyo, Japan) supplemented with or without UltraGRO^TM^-PURE GI GMP grade (hereinafter called “HPL”; AventaCell BioMedical Corp., Atlanta, GA, USA), which is an HPL reagent that has been approved as a material for regenerative medicine products by Pharmaceuticals and Medical Devices Agency in Japan; the PBMCs were then seeded into culture dishes for adherent cells. After 30 min, the non-adherent cells were washed out. The adherent cells were cultured in DCO-K medium that contained 100 ng/mL recombinant human GM-CSF (Miltenyi Biotec B.V. & Co. KG, Bergisch Gladbach, Germany) and 1 µg/mL PEGylated-IFN-α-2b (PEGINTRON^®^; MSD K.K., Tokyo, Japan) with or without 5% HPL for 3 days. The collected immature DCs were suspended in DCO-K medium with or without 5% HPL that contained 100 ng/mL GM-CSF, 1 µg/mL PEGINTRON^®^, 10 µg/mL OK-432 (Chugai Pharmaceutical Co., Ltd., Tokyo, Japan), 10 ng/mL PEG_2_ (Kyowa Pharma Chemical Co., Ltd., Toyama, Japan), and 20 µg/mL tumor peptides such as Wilms tumor 1 (WT1; PEPTIDE INSTITUTE, Inc., Osaka, Japan) or melanoma antigen recognized by T cells 1 (MART-1; Medical & Biological Laboratories Co., Ltd., Nagoya, Japan). Hereinafter, this medium is referred to as the maturation cocktail. The cells suspended in the maturation cocktail were cultured in low adherent cell culture dishes and incubated for 24 h.

### 2.3. Cellular Morphology Observations

Cells that were adherent on day 1 were observed using a fluorescence microscope EVOS^®^ FL Cell Imaging System (Thermo Fisher Scientific, Inc., Waltham, MA, USA) to observe cellular adherent state with a 10× objective lens magnification. Harvested mature DCs were observed using the all-in-one inverted fluorescence microscope BIOREVO BZ-9000 (KEYENCE CORPORATION., Osaka, Japan) for cellular morphology with a 40x objective lens magnification.

### 2.4. Cell Surface Marker Analysis

The following fluorochrome-conjugated mouse IgG anti-human monoclonal antibodies (mAbs) were used for the immunofluorescent staining: phycoerythrin (PE)-conjugated anti-CD11c mAbs (BioLegend, Inc., San Diego, CA, USA), fluorescein isothiocyanate (FITC)-conjugated anti-CD14 mAbs (eBioscience, Inc., San Diego, CA, USA), FITC-conjugated anti-CD40 mAbs (eBioscience, Inc.), PE-conjugated anti-CD56 mAbs (Beckman coulter, Inc., Brea, PA, USA), FITC-conjugated anti-CD80 mAbs (BD Biosciences, San Jase, CA, USA), allophycocyanin (APC)-conjugated anti-CD83 mAbs (BioLegend, Inc.), PE-conjugated anti-CD86 mAbs (eBioscience, Inc.), PE-conjugated anti-CD197 (CCR7) mAbs (Research and Diagnostic Systems, Inc., Minneapolis, MN, USA), FITC-conjugated anti-HLA-ABC mAbs (BD Biosciences), PE-conjugated anti-HLA-DR mAbs (eBioscience, Inc.), FITC-conjugated anti-PD-L1 (BD Biosciences), and PE-conjugated anti-PD-L2 (BD Biosciences). FcR blocking reagent (Miltenyi Biotec B.V. & Co. KG) was added to the cells to block the non-specific binding of antibodies to DCs, and the cells were incubated for 10 min at room temperature. Antibodies were added to the cells, followed by 30 min of incubation at 4 °C in the dark. Then, they were washed with PBS and resuspended in PBS supplemented with 1% FBS and 7-amino-actinomycin D (7-AAD; BD Biosciences) for dead cell staining. The stained cells were analyzed with a BD FACSCalibur^TM^ (BD Biosciences) or BD FACSCanto^TM^II (BD Biosciences). The data were analyzed with the BD CellQuest Pro Software (BD Biosciences) or FlowJo^TM^ Software (BD Biosciences).

### 2.5. Analysis of Endocytic and Proteolytic Activities

A total of 100 µg/mL FITC-Dextran (Molecular Probes, Inc., Eugene, OR, USA) or 10 µg/mL DQ ovalbumin (Molecular Probes, Inc.) was added to the maturation cocktail to measure endocytosis and proteolytic activity, respectively. After maturation, the collected DCs were washed with PBS twice, suspended in PBS supplemented with 1% FBS, and then analyzed by flow cytometry.

### 2.6. CTL Induction In Vitro

Immature DCs generated from HLA-A*02:01 PBMCs as described in 2.2.2. were matured using a maturation cocktail with the addition of 20 µg/mL HLA-A*02:01 MART-1 peptides (ELAGIGILTV; Medical & Biological Laboratories Co., Ltd., Nagoya, Japan). After 24 h, DCs were collected as stimulator cells, divided into aliquots, and cryopreserved. CD8^+^ T cells were separated from HLA-A*02:01-autologous PBMCs using CD8 microbeads (Miltenyi Biotec B.V. & Co. KG, Bergisch Gladbach, Germany) and were applied as responder cells. The stimulator and responder cells were co-cultured in a 96-well U-bottom plate at a ratio of 1:10 in AIM-V medium (Thermo Fisher Scientific, Inc., Waltham, MA, USA) supplemented with 5 ng/mL IL-2 (PeproTech, Inc., Rocky Hill, NJ, USA), 5 ng/mL IL-7 (Research and Diagnostic Systems, Inc.), 10 ng/mL IL-15 (PeproTech, Inc., Rocky Hill, NJ, USA), and 50 µg/mL 2-mercapto-ethanol (Bio-Rad Laboratories, Inc., Hercules, CA, USA) as the stimulation medium. After 3 days of cultivation, AIM-V media supplemented with 10% human AB serum (Biowest, Nuaillé, France) and 50 µg/mL 2-mercapto-ethanol were added as expansion medium. Thereafter, DC stimulation and cell expansion were repeated twice with a 3-day interval. The co-cultured cells were collected 14 days after the first stimulation, and 1 × 10^6^ cells were stained with FITC-conjugated anti-CD8 (Beckman Coulter, Inc., Brea, PA, USA), APC-conjugated anti-CD3 (eBioscience, Inc., San Diego, CA, USA), and PE-conjugated T-Select HLA-A*02:01 MART-1 Tetramer-ELAGIGILTV (Medical & Biological Laboratories Co., Ltd., Nagoya, Japan) to detect the MART-1-specific CTLs. Dead cells were excluded by 7-AAD staining in a flow cytometry analysis.

### 2.7. Detection of Cytokine Production

The collected mature IFN-DCs were seeded at a concentration of 1 × 10^6^ cells/mL, transferred into DCO-K serum-free medium (without HPL), then incubated at 37 °C for 24 h. The culture supernatants were subjected to a Bio-Plex or enzyme-linked immunosorbent assay (ELISA) analysis to quantify the following cytokines: Bio-Plex: IL-6, IL-10, IL-12 (p70), IFN-γ, and TNF-α; ELISA: transforming growth factor (TGF) -β1. Basal cytokine levels were also analyzed. All measurements were performed in duplicate using a Bio-Plex assay kit (Bio-Rad Laboratories, Inc., Hercules, CA, USA) and a Human TGF-beta 1 Quantikine ELISA Kit (Research and Diagnostic Systems, Inc., Minneapolis, MN, USA) according to the protocol of each kit.

### 2.8. Enzyme-Linked Immunosorbent Spot (ELISpot) Assay

The ELISpot assays were performed to examine MART-1-specific IFN-γ production using a human IFN-γ ELISpot PLUS kit (Mabtech AB, Nacka Strand, Sweden) according to the manufacturer’s instructions. Briefly, the cells had been cultured for 21 days that were collected and suspended in AIM-V with 10% FBS and 20 µg/mL HLA-A*02:01 MART-1 peptides (ELAGIGILTV; Medical & Biological Laboratories Co., Ltd., Nagoya, Japan) or 20 µg/mL HLA-A*02:01 HIV gag peptides (SLYNTVATL; Medical & Biological Laboratories Co., Ltd., Nagoya, Japan; used as a negative control) to detect MART-1^+^ CTL induction on day 21. These CTLs were placed in 96-well ELISpot plates precoated with IFN-γ monoclonal antibodies at a concentration of 1 × 10^5^ cells per well. The antibody reactions and staining were performed according to the kit protocol. Then, the emerged spots were counted by an automated ELISpot reader (AID ELISpot Reader Classic ELR 07; Autoimmun Diagnostika GmbH, Strassberg, Germany).

### 2.9. Statistical Analysis

The Wilcoxon signed-rank test was used to compare the differences between IFN-DCs and HPL-IFN-DCs. All statistical analyses were performed using IBM SPSS Statistics software version 24 (IBM Japan, Ltd., Tokyo, Japan). Differences were considered statistically significant at a *p*-value < 0.05.

## 3. Results

### 3.1. Generation of IFN-DCs Using Serum-Free Medium Supplemented with HPL

DCs were produced from the patients’ PBMCs in the absence or presence of HPL as described in 2.2.2. (Figure 1a). After cell adhesion on day 1, microscopic observation indicated that many lymphocyte-like small cells remained in the absence of HPL (Figure 1b, upper left panel). Conversely, there was little contamination of lymphocyte-like small cells in the presence of HPL (Figure 1b, upper right panel). Hereinafter, DCs cultured in the absence or presence of HPL are referred to as “IFN-DCs” or “HPL-IFN-DCs,” respectively. Dendritic-like structures were observed in both IFN-DCs and HPL-IFN-DCs under microscopy (Figure 1b, lower panel). The viability and DC/monocyte recovery rates were significantly higher in HPL-IFN-DCs than in IFN-DCs (Figure 1c; median viability: IFN-DCs, 84.2%; HPL-IFN-DCs, 95.5%; yield: IFN-DCs, 14.1%; HPL-IFN-DCs, 25.4%). Additionally, the flow cytometric analyses indicated a lower rate of lymphocyte contamination in the HPL-IFN-DCs. The adhesion procedure followed by cell culturing demonstrated a far superior DC purity in the presence of HPL than in the absence of HPL (Figure 1c; median purity: IFN-DCs, 83.1%; HPL-IFN-DCs, 99.1%). These results reflect the higher viability, yield, and purity of HPL-IFN-DCs compared with IFN-DCs.

### 3.2. Phenotypic Comparison between IFN-DCs and HPL-IFN-DCs

To determine the effect of HPL on the IFN-DC phenotype, we used flow cytometry to analyze the following cell surface markers that had been reported in previous monocyte-derived DC studies [28,29]: CD11c, CD14, CD40, CD56, CD80, CD83, CD86, CCR7, HLA-ABC, HLA-DR, PD-L1, and PD-L2 (Figure 2). Notably, compared with IFN-DCs, HPL-IFN-DCs expressed higher levels of CD11c (median percentage of positive cells: 98.3% in IFN-DCs; 99.9% in HPL-IFN-DCs), CD14 (35.8% in IFN-DCs; 83.6% in HPL-IFN-DCs), CD56 (37.6% in IFN-DCs; 68.4% in HPL-IFN-DCs), and CCR7 (10.3% in IFN-DCs; 37.8% in HPL-IFN-DCs). Conversely, HPL-IFN-DCs displayed significantly decreased expressions of the HLA co-stimulatory molecules CD40 (98.6% in IFN-DCs; 66.9% in HPL-IFN-DCs) and CD80 (84.0% in IFN-DCs; 33.1% in HPL-IFN-DCs) and the maturation marker CD83 (86.8% in IFN-DCs; 64.2% in HPL-IFN-DCs). HLA-DR expression, which is responsible for antigen presentation to helper T cells, was also slightly down-regulated in HPL-IFN-DCs (99.8% in IFN-DCs; 92.7% in HPL-IFN-DCs). There were no significant differences between IFN-DCs and HPL-IFN-DCs regarding the expressions of CD86, a co-stimulatory molecule important for T cell activation (99.6% in IFN-DCs; 99.6% in HPL-IFN-DCs); HLA-ABC, an antigen presentation to killer T cells (100% in IFN-DCs; 100% in HPL-IFN-DCs); and PD-L1 and PD-L2, immune checkpoint molecules (PD-L1^+^: 73.6% in IFN-DCs; 94.6% in HPL-IFN-DCs; and PD-L2^+^: 13.9% in IFN-DCs; 23.5% in HPL-IFN-DCs).

### 3.3. HPL-IFN-DCs Are Highly Capable of Inducing Antigen-Specific CTLs

To investigate the antigen-presenting abilities of generated DCs to CD8^+^ T cells, antigen-specific CTLs were sensitized with IFN-DCs using a MART-1 peptide. MART-1-specific CTLs could be detected by a MART-1 tetramer analysis on days 14 and 21. HPL-IFN-DCs showed significantly higher MART-1-specific CTL induction than IFN-DCs on day 14 (Figure 3a,b; median MART-1 tetramer^+^ CTLs on day 14: IFN-DCs, 0.29%; HPL-IFN-DCs, 2.88%; *n* = 6). Surprisingly, many more MART-1 tetramer^+^ CTLs were found on day 21 in vitro (Figure 3b; median percentage MART-1 tetramer^+^ CTLs on day 21: IFN-DCs, 1.36%; HPL-IFN-DCs, 9.33%). The average numbers of MART-1 tetramer^+^ CTLs had time-dependently increased on day 21 compared with that on day 14, which is demonstrated in the growth curve in Figure 3c. The number of MART-1 tetramer^+^ CTLs increased significantly in both IFN-DCs and HPL-IFN-DCs, from 9.33 × 10^4^ ± 1.07 × 10^5^ and 4.41 × 10^5^ ± 4.72 × 10^5^ cells on day 14, respectively, to 5.69 × 10^5^ ± 5.76 × 10^5^ cells (IFN-DCs) and 1.62 × 10^6^ ± 6.42 × 10^5^ cells (HPL-IFN-DCs) on day 21 (*n* = 6). This quantitative analysis indicates that HPL-IFN-DCs had a significantly greater potential to present antigens to CD8^+^ T cells than IFN-DCs.

### 3.4. CTLs Induced by IFN-DCs or HPL-IFN-DCs Produced IFN-γ in Response to MART-1 Peptides

Although HPL-IFN-DCs significantly induced CTLs, it remains unclear whether the sensitized CTLs would be functional. IFN-γ production by CTLs is crucial for the efficacy of the DC vaccine. The frequency of IFN-γ production from CTLs was calculated in a MART-1-specific manner using ELISpot assays. IFN-γ spots by MART-1 peptide could be specifically detected in T cells co-cultured with either IFN-DCs or HPL-IFN-DCs (Figure 4a; upper panels). Interestingly, there were more IFN-γ-positive spots in CD8^+^ T cells co-cultured with HPL-IFN-DCs (median number of spots: 505.8) than in those co-cultured with IFN-DCs (304.0). This result was consistent with the trend in CTL induction shown in Figure 3b and Figure 4b (*n* = 6). There was no significant difference in the number of spots between the MART-1 peptide (23.8) and the negative control HIV peptide (25.0) in CD8^+^ T cells cultured alone. Moreover, both CD8^+^ T cells co-cultured with IFN-DCs and those co-cultured with HPL-IFN-DCs showed little responsiveness to HIV peptides, indicating that these cells produced IFN-γ in a MART-1-specific manner (12.3 in IFN-DCs; 37.0 in HPL-IFN-DCs).

### 3.5. HPL Upregulated the Endocytic and Proteolytic Activities of IFN-DCs

Figure 3 and Figure 4 show that HPL-IFN-DCs had the ability to induce strong antigen-specific and functional CTLs. To confirm whether the addition of HPL alters functions other than CTL induction capacity, we investigated endocytic and proteolytic activities, which are important functions of APCs. FITC-dextran or DQ ovalbumin was added to the maturation cocktail and were detected by flow cytometry. Interestingly, the Δ median fluorescent intensity (ΔMFI) values of FITC-Dextran (17.1 in IFN-DCs; 68.0 in HPL-IFN-DCs) and DQ ovalbumin (270.9 in IFN-DCs; 589.7 in HPL-IFN-DCs) were significantly higher in HPL-IFN-DCs than in IFN-DCs, as shown in Figure 5.

### 3.6. Compared with IFN-DCs, HPL-IFN-DCs Showed Higher IL-6, IL-10, and TNF-α Production Levels

To evaluate the ability of HPL-IFN-DCs to produce the cytokines involved in antigen presentation, the IL-6, IL-10, IL-12 (p70), IFN-γ, TNF-α, and TGF-β1 levels were measured with a multiplex or ELISA (Figure 6). To measure the cytokines produced from DCs under steady conditions without HPL interference, the prepared IFN-DCs and HPL-IFN-DCs were incubated in DCO-K medium only. DCO-K medium with or without 5% HPL were confirmed to contain undetectable cytokine levels (each cytokine level indicated less than 1 pg/mL in both medium conditions, besides TGF-β1, 3869.5 pg/mL in DCO-K with 5% HPL); therefore, we performed incubation with DCO-K medium only to exclude its effect. Compared with IFN-DCs, HPL-IFN-DCs secreted more IL-6 (median amount: 302.3 pg/mL in IFN-DCs; 2883.0 pg/mL in HPL-IFN-DCs), IL-10 (11.47 pg/mL in IFN-DCs; 132.7 pg/mL in HPL-IFN-DCs), and TNF-α (412.5 pg/mL in IFN-DCs; 1144.4 pg/mL in HPL-IFN-DCs). A very low amount of IL-12 (p70) was detected from the IFN-DCs and HPL-IFN-DCs (median amount: 1.1 pg/mL in IFN-DCs; 0.18 pg/mL in HPL-IFN-DCs); this difference was significant. Conversely, very low levels of both IFN-γ and TGF-β1 were detected in IFN-DCs and HPL-IFN-DCs with no significant difference between the two types of DCs (median IFN-γ level: 0.59 pg/mL in IFN-DCs; 0.38 pg/mL in HPL-IFN-DCs; median TGF-β1 level: 8.02 pg/mL in IFN-DCs; 9.38 pg/mL in HPL-IFN-DCs).

## 4. Discussion

This study showed that HPL could potentiate IFN-DCs with enhanced antigen presentation, endocytosis, and proteolysis during differentiation and maturation from monocytes; this procedure resulted in high viability, yield, and purity.

For the initial validation of the additives, IFN-DCs were cultured in DCO-K medium, DCO-K with 5% human AB serum, or DCO-K with 5% HPL (Appendix A). The viability and yield of IFN-DCs in DCO-K with human AB serum were the lowest among the groups (Appendix A). PDGF, which is more abundant in HPL than in human AB serum, has been reported to inhibit apoptosis and is presumed to contribute to increased viability [30,31]. Moreover, IFN-DCs with AB serum were not found to have a remarkable change in surface markers in HPL-IFN-DCs when compared with IFN-DCs with DCO-K-only (Appendix A). These analyses lead to focusing on HPL-IFN-DCs with unique surface markers, high survival, and yield.

To determine the optimal HPL ratio in the medium, concentrations of 1%, 5%, and 10% were compared with the non-additional control. Despite a few differences, 5% was adopted as the optimal HPL content ratio for further study based on the DC/monocyte yield, cell viability, and the expressions of representative surface markers, such as CD86 and HLA-ABC (Appendix A). Because HPL contains factors that promote monocyte adhesion, such as ICAM-1 and VCAM-1 [25,32], HPL appears to effectively promote the adhesion of monocytes to cell culture dishes, consequently reducing the rate of non-adhesive lymphocyte contamination (Figure 1c). Moreover, apoptosis is inhibited by exosomes contained in platelet-rich plasma and the raw material of HPL [33]; therefore, a similar effect of HPL may lead to a high viability with an increase in the yield of HPL-IFN-DCs (Figure 1c). The revised manufacturing protocol using HPL would be expected to reduce the number of monocytes collected from patients; thus, highly valuable vaccine products would achieve clinical efficiency for cancer vaccination.

As shown in Figure 2, IFN-DCs cultured in serum-free medium showed similar cell surface markers as previously reported on IFN-α-induced DCs [18]. However, HPL-IFN-DCs displayed a unique phenotype with a conservative expression of CD14, a monocyte marker, and lower expressions of DC maturation markers, such as CD40, CD80, and CD83; this phenotype appears to be similar to that of so-called semi-mature DCs [34]. Additionally, similar to semi-mature DCs, HPL-IFN-DCs also displayed cytokine profiles with higher levels of IL-6 and IL-10 (Figure 6). Semi-mature DCs are known to cause immune tolerance [35]. However, unlike semi-mature DCs, HPL-IFN-DCs displayed high CD86 and HLA-DR expression levels and only produced small amounts of TGF-β1. Since immature DCs spontaneously mature and migrate into the lymph nodes after administration, as reported in rhesus monkey and chimpanzee studies [36,37], the degree of maturation of DCs likely depends on the surrounding conditions. After intradermal injection into rhesus monkeys, immature DCs completely migrated out of the skin, whereas mature DCs tended to remain in the dermis [36]. Although it is difficult to determine the maturation degree based only on the phenotyping of HPL-IFN-DCs, the adoption of maturity as a release criterion for a DC vaccine is controversial. CCR7, which is required for DC migration to the lymph nodes, was hardly expressed by conventional IFN-DCs [38]; however, its expression was enhanced in HPL-IFN-DCs (Figure 2). Based on in vitro chemotaxis experiments using a transwell system to examine DCs migration toward CCL19, CCR7 expression plays a critical role in the migration of DCs [39]. This phenotype is expected to increase the rate of migration to the lymph nodes because of the intradermal administration of HPL-IFN-DCs. Additionally, HPL-IFN-DCs showed higher expression of the natural killer cell marker CD56 (Figure 2). Although conventional IFN-DCs partially expressing CD56 resembled CD56^+^ IFN-DCs with cytotoxic activity [16,18], this activity was not found in HPL-IFN-DCs (Appendix A). Conversely, high PD-L1 expression, an immune checkpoint molecule, was detected in both IFN-DCs and HPL-IFN-DCs (Figure 2). PD-L1 is generally known to act suppressively on T cell activation [40]. Furthermore, PD-L1 expression is upregulated during antigen uptake in cDC1 [41], and PD-L expression on DCs is involved in promoting CD4^+^CD25^+^FoxP3^+^ regulatory T cell expansion and function [42]. Ex-vivo-generated HPL-IFN-DCs can be furnished with tumor antigens exploiting their antigen-specific qualities in Figure 3; therefore, it is possible that HPL-IFN-DCs expressing PD-L1 would improve potency to promote cytotoxic and long-lasting immunity under immunosuppressive regulation.

The most remarkable result in our study was that HPL-IFN-DCs showed potent antigen-specific CTL induction abilities (Figure 3). Han P. et al. recently reported that P-selectin in platelets affects P-selectin glycoprotein ligand 1 (PSGL1) in monocytes, and co-culturing of monocytes and platelets enhances cross-presentation abilities [43]. Although it is not clear whether P-selectin is present in HPL, this report supports the evidence that HPL-IFN-DCs would have a high antigen-specific CTL induction capacity. Additionally, we speculate that this high antigen-presenting capacity is due to an enhanced ability to take up and process antigens. The high endocytic and proteolytic capacities of HPL-IFN-DCs (Figure 5) would likely be enhanced by abundant cytokines contained in HPL. Low concentrations of TNF-α and GM-CSF act synergistically in macrophages, leading to a dramatic upregulation of endocytic activity [44]. In addition, it is speculated that epidermal growth factor (EGF) contained in HPL, which is known to stimulate EGF receptor on DCs, would promote endocytosis [45]. Because the experiments on antigen presentation, antigen uptake, and processing are independent analyses, we did not prove a sequential cross-presentation cascade. Future validation is needed to determine the cross-presentation ability in HPL-IFN-DCs. Cytokines produced by HPL-IFN-DCs may also support the antigen-presenting capability. As shown in Figure 6, HPL-IFN-DCs produced higher levels of IL-6, IL-10, and TNF-α than IFN-DCs. IL-6 is required in the early stages of CTL induction, and proliferation is subsequently promoted by IL-2 [46]. IL-10 acts on both sides, functioning as both an immunosuppressive agent [47] and as a direct enhancer for the proliferation and cytotoxic function of human papillomavirus (HPV)-specific CD8^+^ T cells with an increase in intracellular perforin levels [48]. IL-10 injection in mice was shown to enhance anti-tumor immunity immediately after DC vaccine administration [49]. These reports indicate that IL-10 actively plays an important role against acquired immunity dependent on the optimized condition rather than merely regulating via suppression. Elevated TNF-α levels (Figure 6) have a potential role to improve IL-2 responsiveness and promote T cell proliferation [50]. Our results indicated that IL-12 (p70) production is lower in both IFN-DCs and HPL IFN-DCs (Figure 6). Increased IL-12 production by DCs is an important step in the process of CTL priming, and IL-12 is required for the maximal expression of IFN-γ-secreting CTLs in vitro [51]. Our finding indicated that HPL-IFN-DCs could induce a high level of IFN-γ-producing CTLs despite the low production of IL-12, suggesting that CTLs can be activated in a pathway independent of IL-12 (Figure 4). Conversely, both HPL-IFN-DCs and IFN-DCs displayed lower levels of TGF-β1, which downregulates the antigen presentation function [52].

One of the remaining questions is whether HPL-IFN-DCs are close to any DC subset or monocyte-derived cell population in vivo. CD14-positive DCs have been identified in dermis-expressing markers such as DC-SIGN and CX3CR1 [5,53]. Interestingly, these markers were slightly upregulated in HPL-IFN-DCs compared with IFN-DCs (Appendix A). The phenotype of HPL-IFN-DCs is more similar to that of CD14^+^ DCs differentiated from CD34^+^ hematopoietic stem cells than that of dermal CD14^+^ DCs in vivo [54]. Nevertheless, HPL-IFN-DCs displayed a higher capability of inducing differentiation into potent CTL effectors than both Langerhans cells and CD14^+^ DCs. Exposure to HPL markedly increased the expression of CD141 on HPL-IFN-DCs (Appendix A); this marker is characteristically expressed on cDC1 with a high antigen-presenting capacity. Since XCR1 and CLEC9A are also expressed together in these subsets in human blood [4], the expressions of XCR1 and CLEC9A were also examined in HPL-IFN-DCs but were not detected (Appendix A). Because gene expression differs at the single-cell level even in a population of the same phenotype, the ability to determine the exact subtype based solely on surface antigen analysis is limited. HPL components might affect monocytes that differentiate into a novel type of *DCs* mimicking CD14^+^ DCs; therefore, a comprehensive analysis of mRNA expression per single cell can be used to determine the single-cell RNA sequence of HPL-IFN-DCs to understand the differentiation genealogy of the monocyte and functional pathways upregulated by HPL. Such analyses of HPL-IFN-DCs are expected to deepen the understanding of adaptive immunity and contribute to the development of DC vaccination against refractory cancers and infectious diseases to improve human healthcare.

## 5. Conclusions

In the presence of HPL, IFN-DCs displayed higher viability, yield, and purity, showing a unique phenotype with increased expressions of CD14, CD56, and CCR7. HPL-IFN-DCs induced an extremely high level of antigen-specific CTLs that secreted high levels of IFN-γ. Additionally, antigen endocytosis and proteolytic activities were observed in HPL-IFN-DCs. Our findings suggest that ex-vivo-generated HPL-IFN-DCs would exhibit a novel monocyte-derived DC type with high endocytic and proteolytic activities, which provides insights into a unique type of monocyte-derived DCs exposed to HPL. Further gene expression levels in HPL-IFN-DCs must be analyzed to determine the genealogy of the monocyte lineage for future vaccination therapies.

## 6. Patents

S.S. and T.K. are inventors of the patent for manufacturing an IFN-DC vaccine (WO2016/148179).

## Figures and Tables

**Figure 1 vaccines-09-00010-f001:**
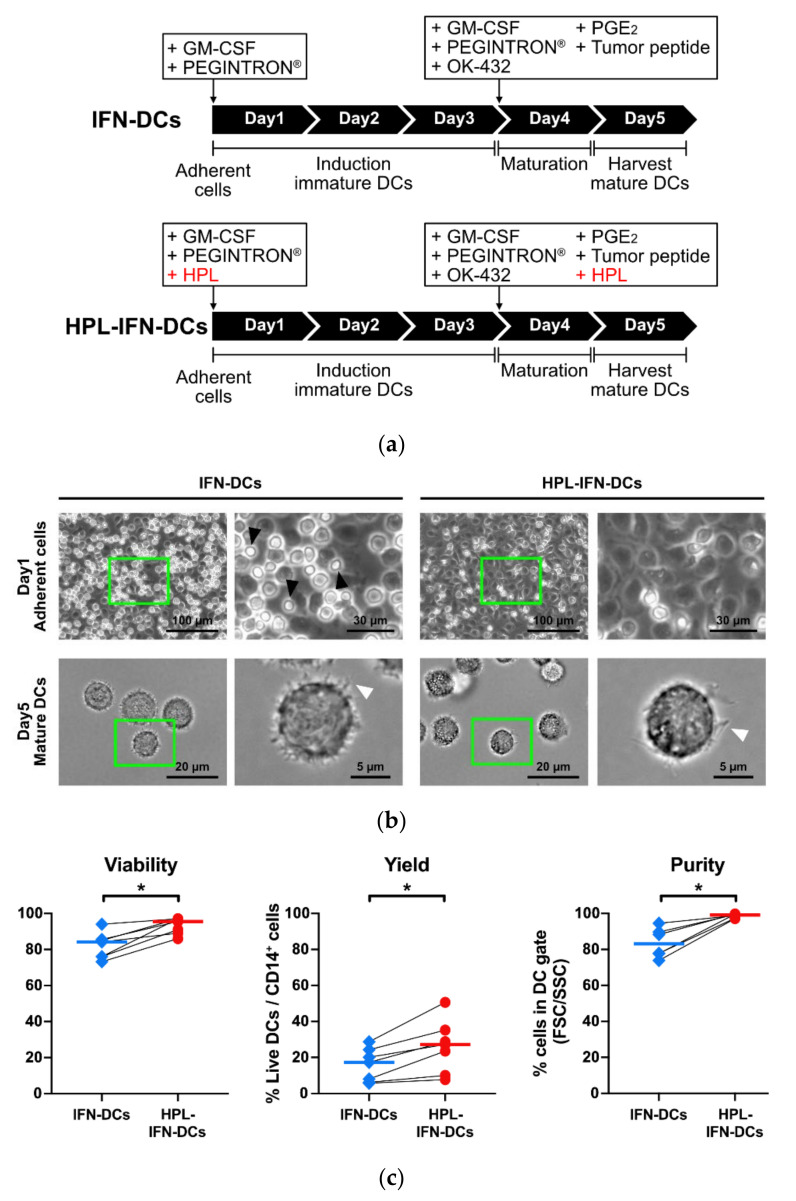
The viability, yield, and purity in IFN-DCs and HPL-IFN-DCs. (**a**) Adherent cells were cultured in DCO-K medium supplemented with or without 5% HPL, as described in the Materials and Methods. In the presence of the HPL conditioning regimen, HPL was continuously added through monocyte selection and harvested DCs. (**b**) The upper panels display micrographs of cells seeded on adherent dishes (magnification: ×10, scale bar: 100 and 30 µm). The black arrowheads indicate small cells, such as lymphocytes. The lower panels display micrographs of the harvested mature DCs in a glass bottom dish (magnification: ×40, scale bar: 20 and 5 µm). The white arrowheads indicate dendrite-like structures. A partial close-up is shown to the right of each photo (green squares). (**c**) Dead cells were measured by trypan blue staining to compare the viability and yield of the DC/monocyte ratio. DC purity was measured by flow cytometry. The gated cells from FSC and SSC, excluding the lymphocyte fraction, were defined as DCs (viability and yield, *n* = 7; purity, *n* = 6). The bold horizontal bars in the graphs indicate the median of each parameter. * *p* < 0.05.

**Figure 2 vaccines-09-00010-f002:**
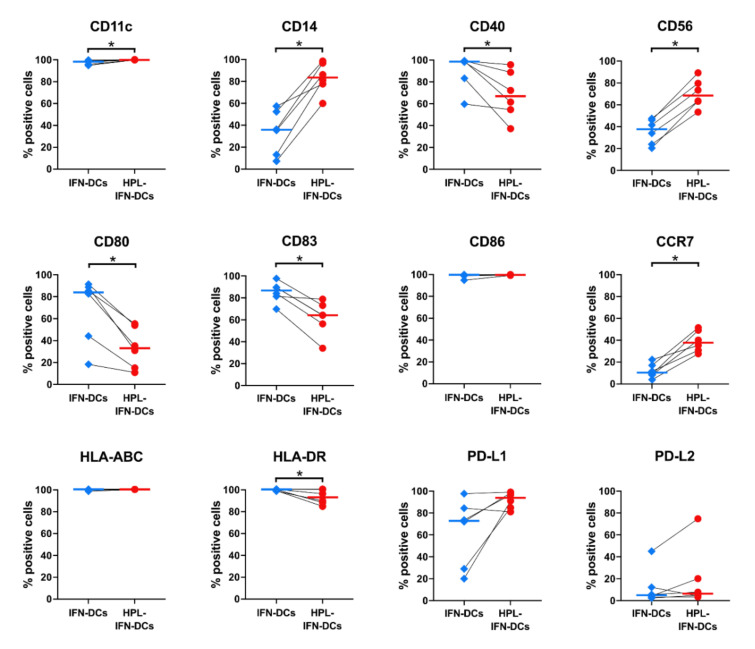
Phenotypic comparison between IFN-DCs and HPL-IFN-DCs. After harvesting IFN-DCs and HPL-IFN-DCs prepared from the same donors, the DCs were stained with antibodies for DC markers and analyzed using flow cytometry (*n* = 6). The bold horizontal bars in the graphs indicate the median of each parameter. * *p* < 0.05.

**Figure 3 vaccines-09-00010-f003:**
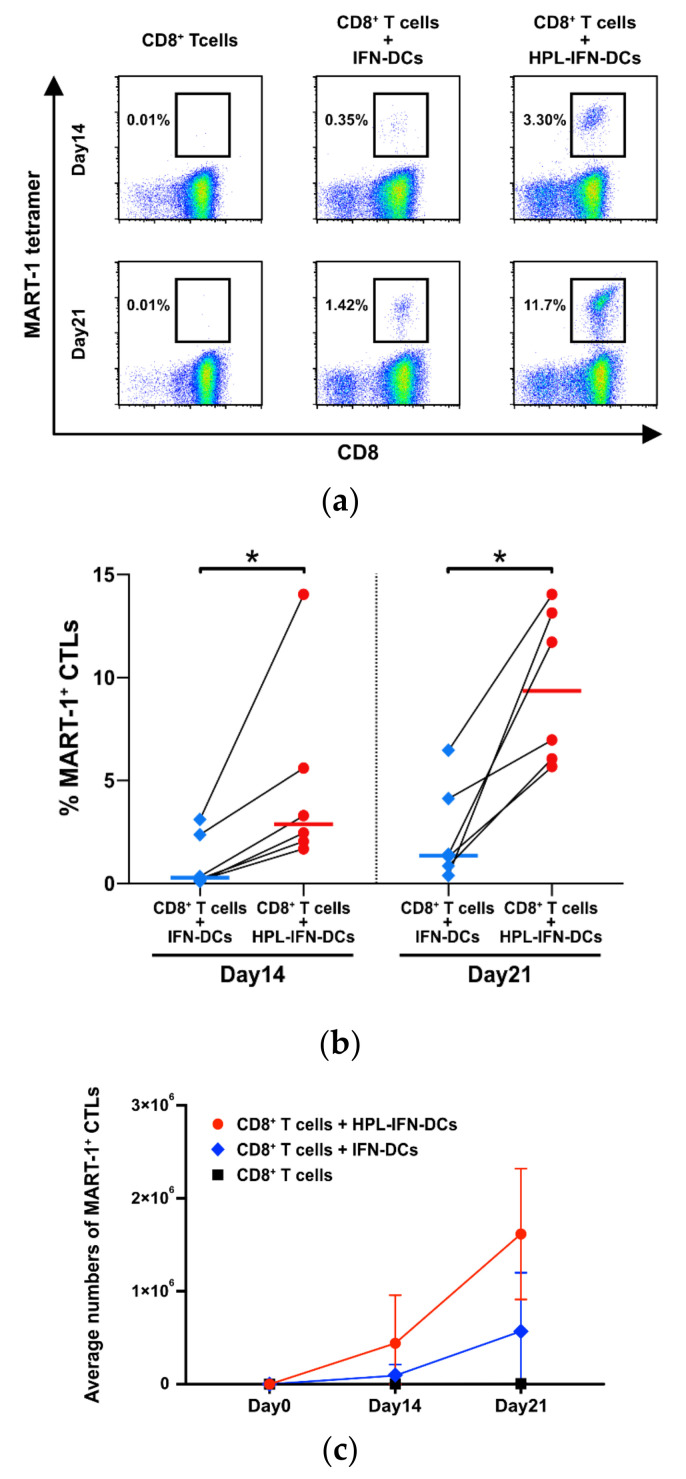
HPL-IFN-DCs for MART-1-specific CTL induction. IFN-DCs or HPL-IFN-DCs were co-cultured with autologous T cells at a ratio of E:T = 1:10. (**a**) Fourteen or 21 days after the start of co-culturing, MART-1-specific CTLs were detected by CD3, CD8, and MART-1 positive gates via flow cytometry (*n* = 6). These dot plots show a representative example. The percentages in the panels indicate the MART-1 tetramer^+^ ratio in CD8^+^ T cells. (**b**) The graph shows the ratio of MART-1 CTLs co-cultured with either IFN-DCs or HPL-IFN-DCs on days 14 and 21. The bold horizontal bars indicate the median of each parameter (*n* = 6). * *p* < 0.05. (**c**) This line graph shows the number of MART-1^+^ CTLs in the culture period (mean ± standard deviation). The vertical axis represents the average number of cells (*n* = 6).

**Figure 4 vaccines-09-00010-f004:**
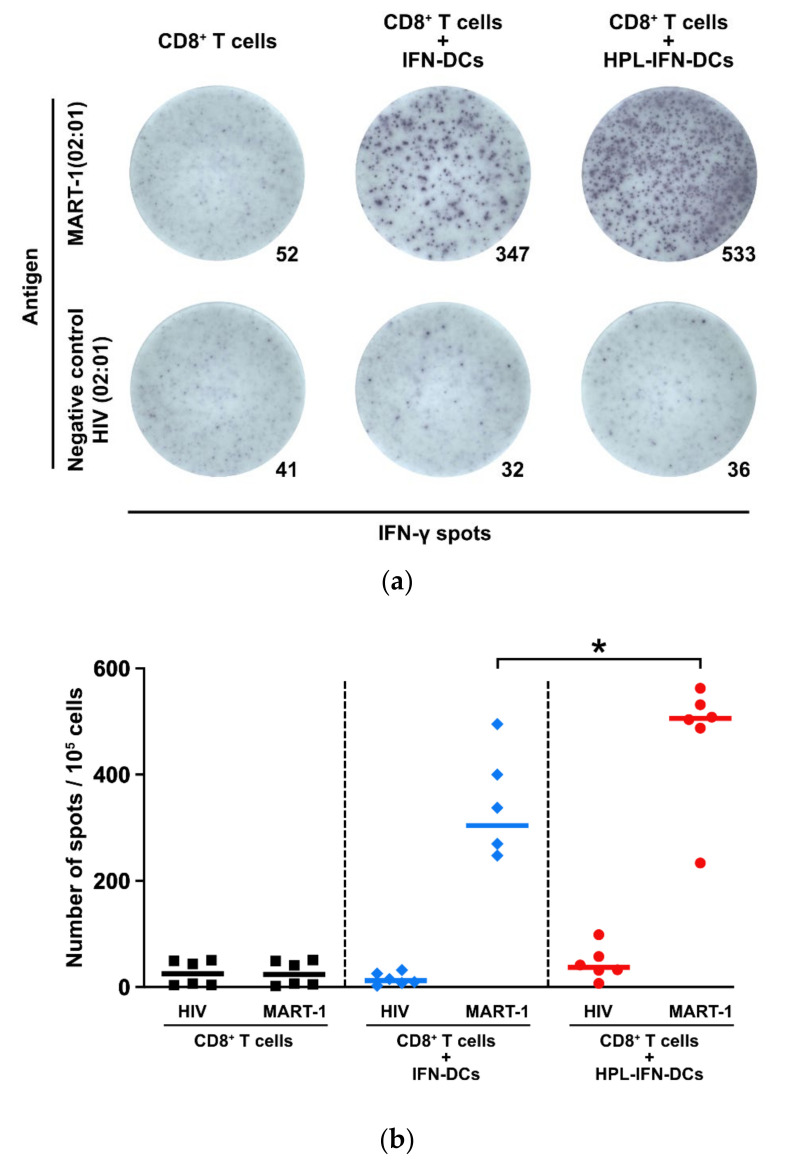
IFN-γ production in response to MART-1 peptide in CTLs. (**a**) The representative ELISpot assays highlight IFN-γ-specific spots upon CTL stimulation with MART-1 peptides (ELAGIGILTV) or HIV peptides as the negative control (*n* = 6). (**b**) The scatter plots indicate the number of spots for each well. The bold horizontal bars in the graphs indicate the median of each parameter. * *p* < 0.05.

**Figure 5 vaccines-09-00010-f005:**
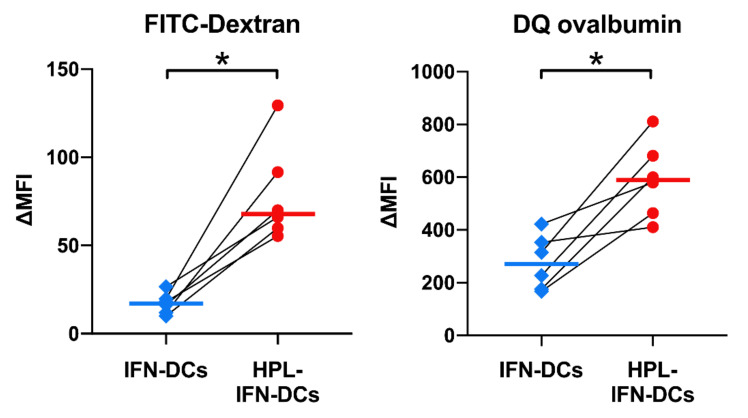
Endocytic and proteolytic activity of HPL-IFN-DCs and IFN-DCs. DCs were incubated with FITC-Dextran to measure antigen endocytosis or DQ ovalbumin to measure proteolytic activity in the maturation cocktail at the time of maturation. These cells were washed after incubating for 24 h, and the fluorescence intensity was measured with flow cytometry (*n* = 6). Δ median fluorescence intensity (ΔMFI) was obtained after subtracting the control incubated with DMSO. The bold horizontal bars in the graphs show the median of each parameter. * *p* < 0.05.

**Figure 6 vaccines-09-00010-f006:**
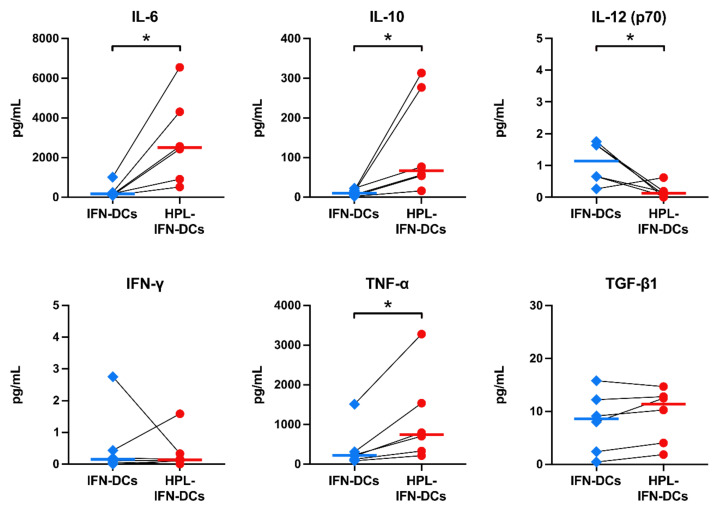
Levels of cytokine production of HPL-IFN-DCs compared to IFN-DCs. Cytokine production was measured in the supernatants collected from IFN-DCs or HPL-IFN-DCs in DCO-K serum-free medium for 24 h. The IL-6, IL-10, IL-12 (p70), IFN-γ, and TNF-α levels were determined with a Bio-Plex multiplex assay, and the amount of TGF-β1 was determined by an ELISA (*n* = 6). The bold horizontal bars in the graphs show the median of each parameter. * *p* < 0.05.

## Data Availability

The data presented in this study are available in the article or Appendix A.

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
