# Peer review of "Interferon-α-Induced Dendritic Cells Generated with Human Platelet Lysate Exhibit Elevated Antigen Presenting Ability to Cytotoxic T Lymphocytes"

_vaccines, 2020, doi:10.3390/vaccines9010010_

Round 1

Reviewer 1 Report

The manuscript describes a novel method to prepare DCs with particular emphasis on using human platelet lysate for improving the various aspect of DCs functionality. The study includes a comparison of viability, yield, purity, cell-surface marker upregulation, cytokine secretion, antigen uptake, processing, CTL priming, their subsequent activation, and toxicity. The discussion on all the specific markers and cytokines along with their implications on the further application is considered to be a significant strength of the manuscript. The data is presented in a very concise and coherent manner. Overall, the study is designed in a methodical manner and the manuscript is well written.

Major comments

  1. Details of viability test and assessment of DCs purity need to be mentioned. Mentioning both trypan blue and 7AAD in the caption of Figure 1 is confusing.
  2. For Figure 3c, please mention the standard deviation for each measurement, and in the test it would be good to mention the values along with range (±).
  3. While the study is comprehensive missing out on migration capacity to lymph node is weakening the claim on CCR7 upregulation (Line 363). Although not critical, the manuscript can be improved by adding an in vitro experiment validating the migration towards lymph nodes. Using DCs migrating towards CCL19 in a transwell system (chemotaxis chamber) has been used in the past.

Author Response

Comments and Suggestions for Authors

The manuscript describes a novel method to prepare DCs with particular emphasis on using human platelet lysate for improving the various aspect of DCs functionality. The study includes a comparison of viability, yield, purity, cell-surface marker upregulation, cytokine secretion, antigen uptake, processing, CTL priming, their subsequent activation, and toxicity. The discussion on all the specific markers and cytokines along with their implications on the further application is considered to be a significant strength of the manuscript. The data is presented in a very concise and coherent manner. Overall, the study is designed in a methodical manner and the manuscript is well written.

Answer) Authors really appreciate the Reviewer's kind comments for revision of our article. We would like to reply and revise the article as optimized as possible following the reviewer's suggestion.

Major comments

1. Details of viability test and assessment of DCs purity need to be mentioned. Mentioning both trypan blue and 7AAD in the caption of Figure 1 is confusing.

Answer) Thank you for your comments. We deleted 7AAD to avoid misreading of the caption in Figure 1 in lines 226-228: The gated cells from FSC and SSC, excluding the lymphocyte fraction, were defined as DCs (viability and yield, n = 7; purity, n = 6).  

2. For Figure 3c, please mention the standard deviation for each measurement, and in the test it would be good to mention the values along with range (±).

Answer) Thank you for your comments. We revised the in Figure 3c with the values of mean ± standard deviation and mentioned in lines 274-275. We also added the data in the text in lines 262-263 to explain the Figure 3c.

3. While the study is comprehensive missing out on migration capacity to lymph node is weakening the claim on CCR7 upregulation (Line 363). Although not critical, the manuscript can be improved by adding an in vitro experiment validating the migration towards lymph nodes. Using DCs migrating towards CCL19 in a transwell system (chemotaxis chamber) has been used in the past.

Answer) Thank you for your suggestion to understand the meaning of the sentence. We added the sentence and cited the reference in Discussion in lines 373-374 as “Based on in vitro chemotaxis experiments using a transwell system to examine DCs migration toward CCL19, CCR7 expression plays a critical role in the migration of DCs [39]. “

Reviewer 2 Report

The manuscript by Date et al. reported the effects of human platelet lysate (HPL) on Interferon-α-induced dendritic cells. The authors showed that compared to the DCs generated in the presence of IFNa alone, HPL-IFNa-DCs have a higher capacity of inducing MART-1 specific cytotoxic T lymphocytes (CTLs). The HPL-IFNa-DCs are phenotypically different from DCs generated by IFNa alone and show higher endocytic and antigen-processing capacity. The manuscript is concise, and the findings are important. However, there are a few issues that the authors need to address.

Major concerns:

  1. The authors need some forms of specificity control; any random cell lysate will work. The phenotypic analysis and explanations will not change. However, the conclusions the authors made in the discussion section about yield, cell viability, and the expressions of surface markers postulated that these (and some other) observations resulted from some components (known or unknown) present in the HPL. To argue this point, the authors need to make IFNa-induced DCs in the presence of random non-specific protein lysate (or protein cocktails) and show that the effects are specific to HPL.
  2. The authors correctly pointed out that the most interesting and surprising results are that the HPL-IFN-DCs showed potent antigen-specific CTL induction abilities despite lower expressions of costimulatory molecules. Once again, the authors mentioned the possibilities of components present in the HPL. However, the authors also proposed that this high antigen-presenting capacity is due to an enhanced ability to take up and process antigens, a process enhanced by abundant cytokines contained in HPL. The method section clearly states that the authors generated CTLs by incubating the DCs with MART-1 peptides (ELAGIGILTV). If this is indeed peptide and not protein, then antigen processing can not be the explanation as the peptide doesn’t need to be processed. Antigen uptake over such a long time point is unlikely to be different between the different groups of DCs. The authors should comment on and explain this observation.

Minor concerns:

  1. The title is confusing and does not convey the DC-specific effects of HPL.
  2. It will be helpful if the authors could show the basal cytokine levels (or the lack of any cytokine production) shortly after the DCs have been transferred into DCO-K serum-free medium (without HPL). This experiment would make sure that there is no pre-existing contamination from HPL.
  3. The authors should avoid the term “phagocytosis”. The authors measured fluid-phage uptake of a soluble protein (endocytosis) and did not measure phagocytosis of particles.

Author Response

Comments and Suggestions for Authors

The manuscript by Date et al. reported the effects of human platelet lysate (HPL) on Interferon-α-induced dendritic cells. The authors showed that compared to the DCs generated in the presence of IFNα alone, HPL-IFNa-DCs have a higher capacity of inducing MART-1 specific cytotoxic T lymphocytes (CTLs). The HPL-IFNα-DCs are phenotypically different from DCs generated by IFNa alone and show higher endocytic and antigen-processing capacity. The manuscript is concise, and the findings are important. However, there are a few issues that the authors need to address.

Answer) Authors really appreciate the Reviewer's kind comments for revision of our article. We would like to reply and revise the article as optimized as possible following the reviewer's suggestion.

Major concerns:

1. The authors need some forms of specificity control; any random cell lysate will work. The phenotypic analysis and explanations will not change. However, the conclusions the authors made in the discussion section about yield, cell viability, and the expressions of surface markers postulated that these (and some other) observations resulted from some components (known or unknown) present in the HPL. To argue this point, the authors need to make IFNα-induced DCs in the presence of random non-specific protein lysate (or protein cocktails) and show that the effects are specific to HPL.

Answer) Thank you for Reviewer’s critical suggestion. We added Supplement Table S1 and Figure S1 shown in Supplement document to answer the reason to promote further study.  For the initial validation of the additives, IFN-DCs were cultured in DCO-K medium, DCO-K with 5% human AB serum, or DCO-K with 5% HPL (Table S1). The viability and yield of IFN-DCs in DCO-K with human AB serum were applied; however, the yield and viability of IFN-DCs cultured in DCK-K medium + AB serum indicated too low values to adopt further experiments (Table S1). IFN-DCs cultured with HPL displayed remarkable phenotypes as ever seen (Figure S1). Thereafter, we decided to determine the protocol shown in Figure 1a.

We also added the second paragraph with references in Discussion in lines 337-344 for better understanding of the article. We replaced Supplementary Materials in lines 452-456.

2. The authors correctly pointed out that the most interesting and surprising results are that the HPL-IFN-DCs showed potent antigen-specific CTL induction abilities despite lower expressions of costimulatory molecules. Once again, the authors mentioned the possibilities of components present in the HPL. However, the authors also proposed that this high antigen-presenting capacity is due to an enhanced ability to take up and process antigens, a process enhanced by abundant cytokines contained in HPL. The method section clearly states that the authors generated CTLs by incubating the DCs with MART-1 peptides (ELAGIGILTV). If this is indeed peptide and not protein, then antigen processing can not be the explanation as the peptide doesn’t need to be processed. Antigen uptake over such a long time point is unlikely to be different between the different groups of DCs. The authors should comment on and explain this observation.

Answer) Thank you for Reviewer’s suggestion. To answer the reviewer’s comments, we characterized functional profiles applied on HPL-IFN-DCs using independent analyses, providing independent Figures 3, 4, and 5. These data did not explain the hole mechanisms for cross presentation ability in IFN-DCs and HPL-IFN-DCs.

We added the sentences in Discussion in lines 397-401 with deleting a sentence as notation not to say to much based on the results in our experiments. We also replaced the phagocytosis or phagocytic ability with endocytosis as Reviewer’s comments to avoid confused explanation in the abstract, text, and graphic abstract.

Minor concerns:

1. The title is confusing and does not convey the DC-specific effects of HPL.

Answer) Thank you for Reviewer’s comments. Authors could understand Reviewer’s suggestion enough to change the title as Human Platelet Lysate Enhances Interferon-α-induced Dendritic Cells on Antigen Presentation to Cytotoxic T Lymphocytes.

2. It will be helpful if the authors could show the basal cytokine levels (or the lack of any cytokine production) shortly after the DCs have been transferred into DCO-K serum-free medium (without HPL). This experiment would make sure that there is no pre-existing contamination from HPL.

Answer) Thank you for Reviewer’s comments. We added the exact methods: 2.7 Detection of Cytokine Production in lines 177-178 and line 181. The base line values of cytokine data were put in the text in lines 314 to 317 to determine the levels derived from IFN-DCs and HPL-IFN-DCs.

3. The authors should avoid the term “phagocytosis”. The authors measured fluid-phage uptake of a soluble protein (endocytosis) and did not measure phagocytosis of particles.

Answer) As mentioned above, we replaced the term as endocytosis thoroughly.

Round 2

Reviewer 2 Report

A complete explanation of the phenotype (HPL-IFN-DCs showed potent antigen-specific CTL induction abilities despite lower expressions of costimulatory molecules) would have benefitted the manuscript but might be beyond the scope of the paper. The authors did a commendable job in addressing the other questions, though this reviewer feels that the title is still a bit obscure.

Author Response

Comments and Suggestions for Authors

A complete explanation of the phenotype (HPL-IFN-DCs showed potent antigen-specific CTL induction abilities despite lower expressions of costimulatory molecules) would have benefitted the manuscript but might be beyond the scope of the paper. The authors did a commendable job in addressing the other questions, though this reviewer feels that the title is still a bit obscure.

Answer)

Thank you for the Reviewer ‘s deep suggestion for us to have the manuscript being better quality.

Please find the R2 files of Word and PDF cleared with delete lines.

We revised the manuscript following the Reviewer’s suggestion.

We changed the title as our best revision as “Interferon-α-induced Dendritic Cells Generated with Human Platelet Lysate Exhibit Elevated Antigen Presenting Ability to Cytotoxic T Lymphocytes”.

In relation with the scope of the article, we deleted the following words as “Despite having lower CD40, CD80, and CD83 levels,” in lines 24-25 in Abstract, lines 384-385 in Discussion, and lines 435-436 in Conclusion.